# Unified failure model for landslides, rockbursts, glaciers, and volcanoes
Qinghua Lei [1] ✉ & Didier Sornette[2]

Forecasting catastrophic failures that threaten life and property remains a formidable challenge. A major hurdle lies in the intermittent rupture dynamics of heterogeneous materials. This erratic pattern challenges conventional time-to-failure predictive models, which typically assume a smooth, monotonic power law acceleration. Here, we propose a unified failure model based on a log-periodic power law that encapsulates the intermittent acceleration-deceleration sequences within a single framework. We validate this unified model using a global dataset of 109 historical geohazard events including landslides, rockbursts, glacier breakoffs, and volcanic eruptions, spanning a century and across seven continents. We show that our model significantly outperforms the conventional approach, offering a robust and versatile framework for describing the complex rupture behavior of diverse geomaterials such as rock, soil, and ice at the site scale. This unified perspective not only broadens the model's applicability across diverse geohazards but also highlights its potential to enhance early warning systems.

Catastrophic failure occurs in a wide spectrum of geological materials such as rock, soil, and ice, driving various extreme geohazards like landslides, rockbursts, glacier breakoffs, and volcanic eruptions[1–8]. Thus, it is crucial to develop the best possible predictive understanding of these rupture phenomena, which is a fundamental goal of numerous scientific and technological disciplines, including geomechanics, seismology, glaciology, volcanology, and engineering. Various empirical and physical approaches have been proposed to describe geomaterial failure, with the power law singularity (PLS) model and the derived inverse rate technique[9–12] being widely adopted for time-to-failure analysis of geohazard events[3,13–19]. Over the past decades, great efforts have also been devoted to develop and deploy high-precision monitoring technologies to observe various geohazard phenomena[2,3,8,20]. However, only a limited number of catastrophic events have been successfully predicted so far. One major source of uncertainty stems from the sporadic nature of rupture in heterogeneous materials, typically marked by a sequence of progressively shorter quiescent phases interspersed with sudden intense bursts[21–24], rather than a smooth, monotonic progression of deformation and damage. This seemingly erratic pattern complicates failure predictions as it challenges the continuous scale invariance assumed by the simple power law[9–11,13].

Here, we propose a unified failure model based on the log-periodic power law singularity (LPPLS) formulation, simultaneously capturing the accelerating and decelerating phases within a single framework. Amounting mathematically to a generalization of the power law exponent from real to complex numbers (see Methods and Supplementary Notes 1–2), this unified failure model, rooted in statistical physics[25], captures the partial breakdown of the symmetry of continuous scale invariance to discrete scale invariance[26] that is inherent to the non-monotonic and intermittent dynamics of damage and rupture processes in heterogeneous materials. Thus, the LPPLS model leverages the oscillatory patterns in rupture dynamics, transforming them from traditionally perceived nuisances or noises into essential components of the predictive framework. We demonstrate the superiority of this LPPLS-based unified failure model over the PLS-based conventional failure model using an extensive global dataset of historical geohazard events.

## Results

We compile a comprehensive global dataset of 109 catastrophic geohazard events, including landslides, rockbursts, glacier breakoffs, and volcanic eruptions, spanning seven continents over the past century (see Fig. 1, Supplementary Note 3 and Supplementary Tables 1–4). This dataset covers different types of monitoring data, such as geodetic observations, geophysical records, and geochemical measurements (see Supplementary Tables 1–4). Using this global dataset, we examine the applicability and accuracy of the LPPLS-based unified failure model across diverse contexts as well as provide a detailed comparison against the PLS-based conventional failure model.

We implement a stable and robust scheme for calibrating the LPPLS and PLS models against monitoring data (see Supplementary Notes 4–6 for the calibration procedures). We then examine their performance based on a comprehensive set of evaluation metrics (see Methods). More specifically,

[1]Department of Earth Sciences, Uppsala University, Uppsala, Sweden. [2]Institute of Risk Analysis, Prediction and Management, Academy for Advanced Interdisciplinary Studies, Southern University of Science and Technology, Shenzhen, China. ✉e-mail: qinghua.lei@geo.uu.se

**Fig. 1 | Global distribution of 109 historical geo-hazard events including landslides, rockbursts, glacier breakoffs, and volcanic eruptions.** The size of each event is calculated as the cube root of the failure volume.

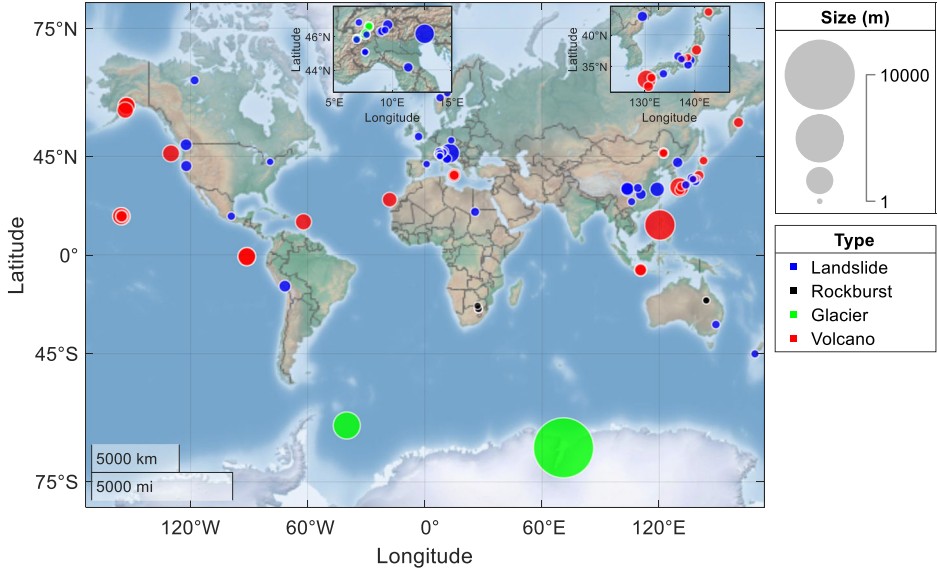

we first analyze the frequency distribution and empirical cumulative distribution function (eCDF) of residuals and further compute the normalized root mean square error (NRMSE). These metrics enable us to quantify the mismatch between the models and data. We also compute the normalized Akaike information criterion (NAIC) and normalized Bayesian information criterion (NBIC), which balance goodness-of-fit with a penalty term for the number of parameters, thereby preventing overfitting. This enables an assessment of the relative quality of the two competing models having different numbers of parameters, with the one having the lower NAIC and NBIC values preferred. Furthermore, we test the null hypothesis $H_0$ that the data follow the PLS model (against the alternative that the data follow the LPPLS model) using the Wilks test, which is the most powerful test for comparing two competing models with one nested in another. If the $p$-value for the Wilks test is below the prescribed significance level (e.g., 0.05), $H_0$ is rejected. Moreover, we perform two-sample Kolmogorov-Smirnov (KS) and Anderson-Darling (AD) tests of the null hypothesis $\mathscr{H}_0$ that the probability distributions of the LPPLS and PLS residuals are identical. If the $p$-value is smaller than the significance level, $\mathscr{H}_0$ is rejected, which may additionally indicate model superiority. It is crucial to emphasize that the model calibration presented here is not a simple statistical fitting exercise; rather, it employs a physically based formula with a rigorous rigid mathematical structure. All the model parameters have well-defined physical meanings and are linked to the underlying processes that govern geomaterial ruptures (see Methods). This approach incorporates fundamental physical principles, capturing the underlying damage and rupture processes occurring in heterogeneous materials (see Methods and Supplementary Notes 1–2). Unlike purely statistical methods such as polynomial regression, which lack grounding in physical insights or constraints, our model explicitly integrates clear physical ingredients, ensuring both robustness and a strong connection to the actual processes at play.

The first study case is the Veslemannen landslide in Norway. This instability complex, consisting of high-grade metamorphic rocks, has been continuously monitored since October 2014 by a ground-based interferometric synthetic-aperture radar system[27] with an accuracy of 0.5 mm. This landslide primarily exhibited active movements during summer and autumn seasons, likely due to rainwater infiltration into the slope through the thawed upper frost zones. On September 5, 2019, ~54,000 m³ rock collapsed. Prior to the failure, this landslide shifted considerably, displacing several meters over the course of ~3 months, showing an accelerating and oscillating behavior (see Fig. 2a for the monitoring data at one of the seven radar points and see Supplementary Fig. 8 for all the radar points). It is evident that the LPPLS model gives a remarkable match to the data, with

both the acceleration and deceleration phases well captured, whereas the PLS model only depicts the general trend (Fig. 2a). The LPPLS residuals are confined in a much narrower range compared to the PLS residuals (Fig. 2a inset). The LPPLS model shows much lower NRMSE, NAIC, and NBIC values, while the $p$-values for the Wilks, KS, and AD tests are all well below 0.05 (Table 1), suggesting that the LPPLS model outperforms the PLS model. The excellent performance of the LPPLS model is further demonstrated by its estimated $t_c$ (critical time of the rupture) which is very close to the actual failure time (with a small discrepancy of 0.13 day; note that the data have a daily aggregation resolution), whereas the $t_c$ estimated by the PLS model is ~9 days after the actual failure (Table 2).

The second example is a rockburst event that occurred in an underground mine at New South Wales, Australia[28]. This coal mine, located at a depth of ~250 m, was mined using the longwall method. An in-situ monitoring system using multipoint extensometers (with an accuracy of 0.5 mm) was installed to record the roof displacement of a gateroad, which is a rectangular tunnel with a width of 5.2 m and a height of 3 m providing access to the longwall face. The roof catastrophically failed on June 4, 2004, prior to which a precursory accelerating and oscillating behavior was observed (Fig. 2b). A visual inspection indicates that the LPPLS model well captures the superimposed acceleration-oscillation behavior of the roof, characterized by alternating accelerating and decelerating phases, whereas the PLS model can only track the overall trend (Fig. 2b). This is consistent with the frequency and eCDF distributions of residuals (Fig. 2b inset), demonstrating that the LPPLS model more closely matches the data. The LPPLS model is also associated with smaller NRMSE, NAIC, and NBIC values than the PLS model, while the $p$-values for the Wilks, KS, and AD tests are all below 0.05 (Table 1). Furthermore, the estimated $t_c$ by the LPPLS model is closer to the actual failure time (Table 2). All these results suggest that the LPPLS-based unified failure model fits accurately to the data and surpasses the conventional PLS model.

The next case is a hanging cold glacier located on the south face of Grandes Jorasses, Italy and at an elevation of 3950 m above sea level[3]. Surface displacements of this glacier, measured by a robotic total station with multiple reflectors, were evaluated with an accuracy of ~1 cm. A large volume of ~105,000 m³ ice broke off[29] in two consecutive events on September 23 and 29, 2014. We focus on the first breakoff event. One can see that this glacier experienced a generally smooth acceleration behavior over several months, for which both the LPPLS and PLS models show a close match to the trend (Fig. 2c) and provide excellent estimates of the failure time (Table 2). However, the LPPLS residuals are much smaller than the PLS residuals (Fig. 2c inset). The NRMSE, NAIC, and NBIC values of the LPPLS

**Fig. 2 | Examination of the LPPLS-based unified failure model in describing the monitoring data of various geohazard events with a comparison to the PLS-based conventional failure model. a** Slope surface displacement (aggregated on a daily basis) prior to a catastrophic landslide at Veslemannen, Norway. **b** Tunnel roof displacement (aggregated on an hourly basis) prior to a violent rockburst at New South Wales, Australia. **c** Glacier surface displacement (aggregated on a daily basis) prior to a rapid ice breakoff at Grandes Jorasses, Italy. **d** Seafloor uplift (aggregated on a weekly basis) prior to a volcanic eruption at Axial Seamount, Pacific Ocean. Note that the raw measurement data for these events, originally recorded at higher frequencies, are aggregated on a lower frequency to expedite calibration and facilitate visualization.

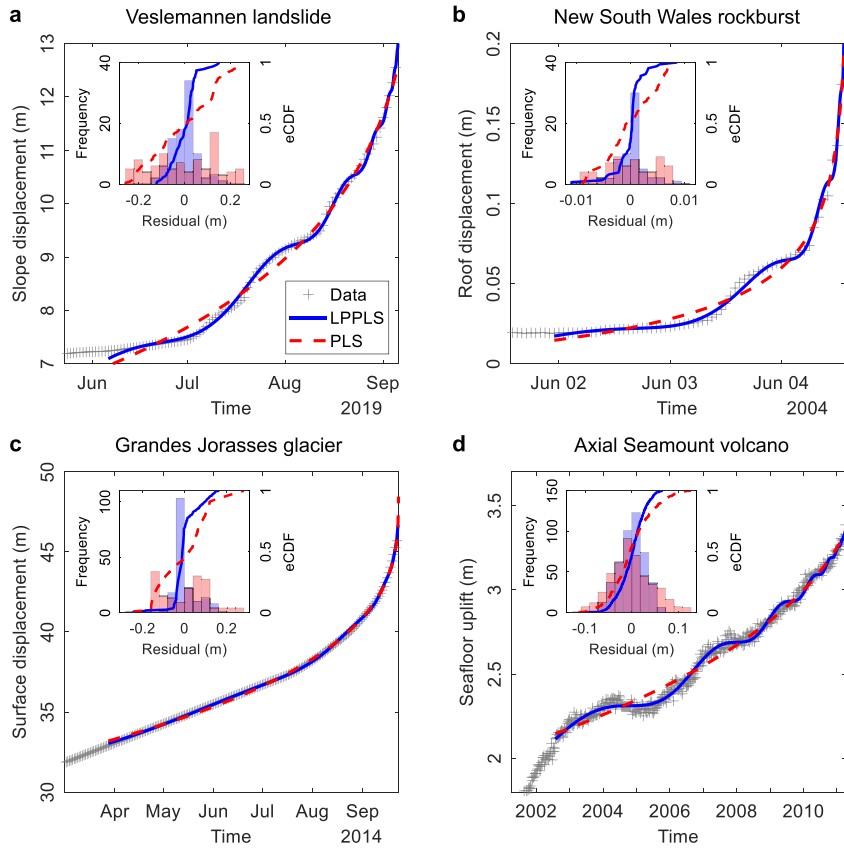

model are considerably lower than those of the PLS model, while the *p*-values for the Wilks, KS, and AD tests are all well below 0.05 (Table 1). These metrics collectively demonstrate that the LPPLS-based unified failure model performs better than the conventional PLS model.

The fourth example is Axial Seamount, an active submarine volcano with a summit caldera at ~1.5 km depth and a base at ~2.4 km, located ~500 km offshore Oregon, USA[30]. This basaltic volcano with magma supplied from the Cobb hotspot has erupted three times over the past 26 years, in 1998, 2011, and 2015. It has been closely monitored by a cabled network of seafloor instruments since 1998, with the seafloor vertical deformation measured at a resolution of ~1 cm through bottom and mobile pressure recorders[31]. We focus on the eruption event in April 2011, prior to which the volcano exhibited a series of inflation-deflation cycles over ~8 years (Fig. 2d). The LPPLS model effectively captures the oscillatory behavior of the seafloor deformation superimposed on an overall acceleration, whereas the PLS model only depicts the overall trend (Fig. 2d). Compared to the PLS model, the LPPLS model is associated with smaller residuals (Fig. 2d inset) and lower values for NRMSE, NAIC, and NBIC (Table 1). Additionally, the *p*-values for the Wilks, KS, and AD tests are all well below 0.05 (Table 1). The failure time estimated by the LPPLS model has a discrepancy of only 6 days compared to the actual eruption time (i.e., just one point of the weekly-aggregated data resolution), while that by the PLS model is almost 1 year after. All these results support the superior performance of our failure model.

We further test the unified failure model for describing the geomaterial failure behavior during various geohazard events (see Fig. 3 for some typical examples among 109 events analyzed in total; refer to Supplementary Figs. 1–14 for the complete inventory). More specifically, Fig. 3a shows the slope displacement data monitored by an extensometer at the Preonzo rockslide, Switzerland[32] which collapsed on May 15, 2012. Figure 3b presents the displacement time series constructed from high-frequency optical satellite images for the Achoma landslide, Peru[33], which failed in June 2020. Figure 3c illustrates the displacement evolution of the Vajont rockslide,

Italy[34], prior to a catastrophic failure on October 9, 1963. Figure 3d shows the acceleration behavior recorded by an extensometer at a quick clay slide at Roesgrenda, Norway[35]. Figure 3e displays the displacement data of a rock cliff acquired from a Terrestrial LiDAR instrument at Puigcercós, Spain[36], where a rockfall event occurred on December 3, 2013. Figure 3f depicts the displacement time series of a subvertical granitic slope monitored by a ground-based synthetic aperture radar at Gallivaggio, Italy[37], where a rockfall event occurred on May 29, 2018. Figure 3g shows the stope closure data measured by a closure meter prior to a violent rockburst in a deep platinum mine, South Africa[38]. Figure 3h gives the cumulative Benioff strain of seismic energy released before a large rockburst event in a deep gold mine, South Africa[39]. Figure 3i shows the multi-year rift propagation in the Amery Ice Shelf, Antarctica[40], preceding a large calving event on September 25, 2019. Figure 3j depicts the temporal evolution of area loss of the UK211 iceberg[41] that rapidly disintegrated in 2006, tracked by the satellite-based Moderate Resolution Imaging Spectroradiometer sensor. Figure 3k displays the surface displacement data of a polythermal glacier at Weissmies, Switzerland[42] prior to a breakoff on September 10, 2017, recorded by a high-resolution camera. Figure 3l gives the displacement data of a polythermal glacier at Planpincieux, Italy[43] prior to a breakoff on September 16, 2015. Figure 3m illustrates the temporal evolution of earthquake count prior to an explosive eruption of the Merapi stratovolcano, Indonesia[44] in October 2010. Figure 3n shows the ground uplift data recorded by a continuous Global Positioning System network during the 2018 eruption of the Sierra Negra shield volcano, Ecuador[45]. Figure 3o plots the tiltmeter measurements at the St. Helens stratovolcano during its 1982 eruption[46]. Figure 3p shows the temporal variation of cumulative normalized soil $CO_2$ efflux recorded by a geochemical monitoring network at the Etna stratovolcano[47] that erupted in September 2013.

The unified failure model provides an excellent fit to the documented failure phenomena in all these cases, effectively capturing the alternating phases of acceleration and deceleration in geological systems as they approach a final breakdown (a detailed comparison with the PLS model is

**Table 1 | Performance evaluation of the LPPLS and PLS models in describing the monitoring data of the Veslemannen landslide, New South Wales rockburst, Grandes Jorasses glacier, and Axial Seamount volcano**

| Evaluation metrics | Veslemannen landslide | New South Wales rockburst | Grandes Jorasses glacier | Axial Seamount volcano |
|---|---|---|---|---|
| $\text{NRMSE}_{\text{LPPLS}}$ | $5.55 \times 10^{-4}$ | $4.34 \times 10^{-5}$ | $2.56 \times 10^{-4}$ | $6.02 \times 10^{-4}$ |
| $\text{NRMSE}_{\text{PLS}}$ | $3.73 \times 10^{-3}$ | $1.42 \times 10^{-4}$ | $1.67 \times 10^{-3}$ | $1.55 \times 10^{-3}$ |
| $\text{NAIC}_{\text{LPPLS}}$ | $-2.84$ | $-8.85$ | $-2.80$ | $-4.31$ |
| $\text{NAIC}_{\text{PLS}}$ | $-1.00$ | $-7.76$ | $-1.46$ | $-3.37$ |
| $\text{NBIC}_{\text{LPPLS}}$ | $-2.64$ | $-8.61$ | $-2.68$ | $-4.24$ |
| $\text{NBIC}_{\text{PLS}}$ | $-0.89$ | $-7.62$ | $-1.39$ | $-3.34$ |
| $p$-value (Wilks test) | 0.00 | 0.00 | 0.00 | 0.00 |
| $p$-value (KS test) | 0.00 | 0.03 | 0.00 | 0.00 |
| $p$-value (AD test) | 0.00 | 0.00 | 0.00 | 0.00 |

**Table 2 | Parameters of the LPPLS and PLS models calibrated to the monitoring data of the Veslemannen landslide, New South Wales rockburst, Grandes Jorasses glacier, and Axial Seamount volcano**

| Model parameters | Veslemannen landslide | New South Wales rockburst | Grandes Jorasses glacier | Axial seamount volcano |
|---|---|---|---|---|
| LPPLS: | | | | |
| $t_c$ | 0.13 | 0.02 | 0.00 | 6.00 |
| $m$ | 0.42 | $-0.12$ | 0.33 | 0.64 |
| $\omega$ | 7.49 | 4.94 | 4.94 | 9.23 |
| $\phi$ | $-0.82$ | $-0.01$ | 0.65 | 0.61 |
| $A$ | 13.64 | $-0.24$ | 48.76 | 3.38 |
| $B$ | $-1.03$ | 0.28 | $-2.88$ | $-0.01$ |
| $C$ | $4.09 \times 10^{-2}$ | $5.77 \times 10^{-3}$ | $3.56 \times 10^{-2}$ | $5.12 \times 10^{-4}$ |
| PLS: | | | | |
| $t_c$ | 9.23 | 0.05 | 0.00 | 323.10 |
| $m$ | 0.07 | $-0.28$ | 0.34 | 0.32 |
| $A$ | 43.93 | $-0.08$ | 48.46 | 4.38 |
| $B$ | $-26.85$ | 0.12 | $-2.66$ | $-0.16$ |

Note: $t_c$ is in day; $m$, $\omega$, and $\varphi$ are dimensionless; $A$ is in meter; $B$ and C are in meter per day$^m$. The actual time of failure corresponds to time $t = 0$ day. The start of the calibration time window is detected by the Lagrange regularization approach (see Supplementary Note S6), while the calibration time window ends at the last available data point before the actual failure occurs

given in Supplementary Tables 9–12. In total, we have tested 109 historical events (consisting of 160 time series data), including 49 landslides, 11 rockbursts, 17 glacier breakoffs, and 32 volcanic eruptions, spanning seven continents over the past century (see Fig. 1, Supplementary Figs. 1–14, and Supplementary Tables 1–16). Figure 4 illustrates the comparative model performance of LPPLS against PLS across the entire dataset (see Supplementary Tables 9–12 for more details). The NRMSE of the LPPLS model is consistently smaller than that of the PLS model across the entire dataset; notably, for 67.5% of the data, the NRMSE of the LPPLS model is less than half that of the PLS model. The NAIC (respectively NBIC) values of the LPPLS model are lower than those of the PLS model by 0.5 unit for 65% (respectively 70%) of the data and by 1 unit for 35% (respectively 40%). Note that a 0.5 to 1 unit decrease in NAIC or NBIC corresponds to a relative likelihood improvement by a factor of 1.65–2.72 times per data point (i.e., it

is 1.65–2.72 times more likely per data point that each of the 109 dataset is explained by the LPPLS model than by the PLS model). For 91% of the data, the $p$-value from the Wilks test is below 0.05 (see Supplementary Tables 9–12), providing strong evidence of the LPPLS model's superiority. The $p$-values of the KS and AD tests are below 0.05 for 42% and 47% of the data, respectively, aligning with the fact that these tests are less powerful than the Wilks test, particularly given the limited data for some events (see Supplementary Tables 13–16). However, the evidence remains strong, as only 8 cases with $p$-values below 0.05 would be expected if the $p$-values were random. All these results collectively indicate the superior performance of the LPPLS-based unified failure model over the PLS-based conventional failure model, because the former can constrain the fit to the bursty oscillating structures in the data while the latter is often poorly defined, particularly in the presence of complex temporal structures.

## Discussion

Our results reveal that the proposed unified failure model can physically and accurately describe the failure behavior of various geomaterials (rock, soil, and ice) under diverse contexts, ranging from landslides and rockbursts to glacier breakoffs and volcanic eruptions. These widespread geohazard phenomena develop over a wide spectrum of time scales (from hours/days to months/years) and length scales (from meters to tens of kilometers) (see Fig. 1, Supplementary Figs. 1–14, and Supplementary Tables 1–4). This failure model provides a unified framework that captures the alternating acceleration-deceleration phases during geomaterial ruptures, driven by a complex interplay of damage and healing processes in heterogeneous geological systems. The broad applicability of this unified failure model highlights the commonality of different failure phenomena, which are mechanistically driven by similar mechanisms involving damage, healing, and frictional processes, interactions between numerous constituent components, as well as final strain localization in heterogeneous materials. Our model not only captures the fundamental physical processes governing geomaterial rupture phenomena but also remains flexible enough for calibration against monitoring data. Each system is inherently unique, with its own heterogeneity, damage and healing history, and external conditions. Our model adapts to these specific characteristics and idiosyncratic patterns, making this adaptability an extraordinary strength of our framework. Log-periodic oscillations generally appear more pronounced in landslides, rockbursts, and volcanic events than in glaciers, likely due to the relatively homogeneous nature of ice materials. However, our quantitative model comparisons and rigorous statistical analyses provide clear evidence that the LPPLS model consistently outperforms the PLS model across all contexts, including glaciers.

Our analyses show that the critical exponent $m$ predominantly falls in the range of $-1$ to 1, for both PLS and LPPLS models (Supplementary Fig. 15). Correspondingly, the nonlinearity exponent $\alpha = 1 + 1/(1 - m)$ (Supplementary Note 1) is predominantly greater than 1.5, with a concentration around 1.7, while spanning a broad range from below 2 to well above 2 (Supplementary Fig. 15). This deviates from the commonly assumed $\alpha = 2$ in the inverse rate method[10–12,14,19,48–50], suggesting that special caution is needed when adopting this assumption in practice. The $\alpha$ value in the range between 1 and 2 may be explained by the dominance of nucleation and propagation of noninteracting subcritical cracks[15,16], while the occurrence of $\alpha > 2$ may be related to the dominance of crack interaction and coalescence characterized by stronger positive feedbacks. In most cases, the $m$ values derived from the LPPLS and PLS models are generally consistent (Supplementary Tables 5–8 and Supplementary Fig. 15). However, some discrepancies are observed, which may stem from the PLS model's vulnerability to biases due to the presence of intermittent oscillations in the empirical data. Variations of $m$ values across different systems may be attributed to differences in heterogeneity conditions. Extensive laboratory studies have shown that heterogeneities associated with crack distribution, pore space, stress field, and/or fault roughness strongly control the precursory acceleration behavior of geomaterials as they approach catastrophic failure[51–56]. One may also observe that the $m$ values derived from monitoring

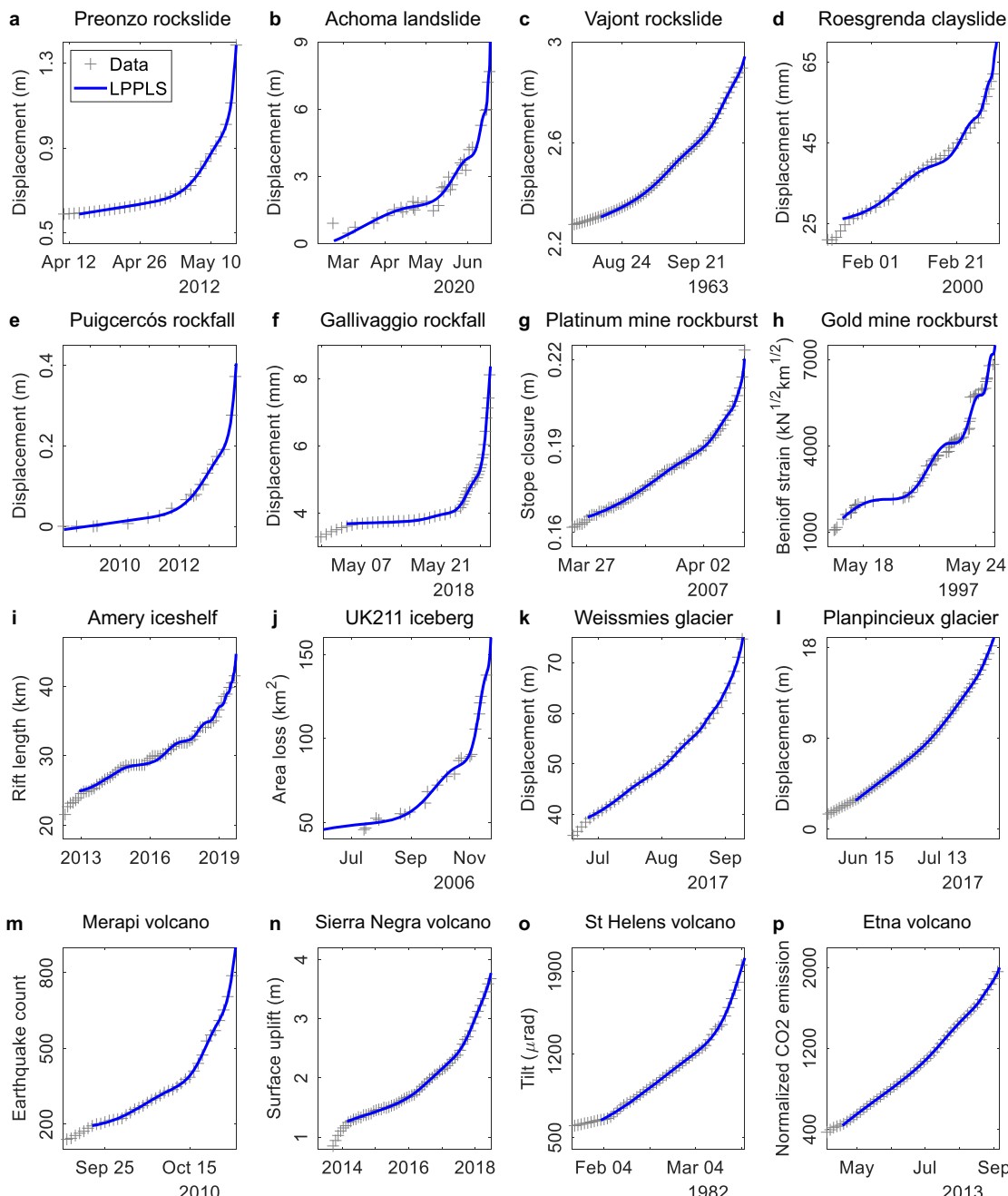

**Fig. 3 | Application of the LPPLS-based unified failure model to various geohazard events. a** Preonzo rockslide, Switzerland. **b** Achoma landslide, Peru. **c** Vajont rockslide, Italy. **d** Roesgrenda clayslide, Norway. **e** Puigcercós rockfall, Spain. **f** Gallivaggio rockfall, Italy. **g** Merensky rockburst, South Africa. **h** Goldmine rockburst, South Africa. **i** Amery iceshelf, Antarctica. **j** UK211 iceberg, Antarctica. **k** Weissmies glacier, Switzerland. **l** Planpincieux glacier, Italy. **m** Merapi volcano, Indonesia. **n** Sierra Negra volcano, Ecuador. **o** St. Helens volcano, USA. **p** Etna volcano, Italy.

data collected by different instruments of the same type at a single site at different points of measurement may vary, reflecting the inherent heterogeneity of natural systems. Variations in $m$ values across different instrument types at the same site arise because each observable measures a different variable within the same system. Indeed, different observables such as strain, earthquake count, and energy release are governed by distinct singularity exponents in their time-to-failure scaling[57]. Additionally, the $m$ values of the same observable for failure events occurring at different times at the same site can change, highlighting the nonstationary nature of natural systems. This is because the system at the onset of each failure cycle may have different initial damage states, which can indeed affect its precursory acceleration pattern, as demonstrated in laboratory experiments[54,56].

In addition to the broad applicability to diverse geohazards, we have also demonstrated that this unified failure model can be applied to various observables (e.g., surface displacement, tunnel closure, energy release, rift length, earthquake count, angular change, and gas emission) recorded by different instruments (e.g., extensometers, reflectors, tiltmeters, closure-meters, satellites, LiDAR, synthetic aperture radar, Global Positioning System, pressure recorders, and seismic/geochemical monitoring networks) (see Supplementary Tables 1–4). Although these observables differ in form and are recorded using a variety of instruments, they are all physically linked to rupture processes in heterogeneous systems, which explains why they can be consistently represented within the same unified modeling framework. These results indicate that this unified failure model is general and robust,

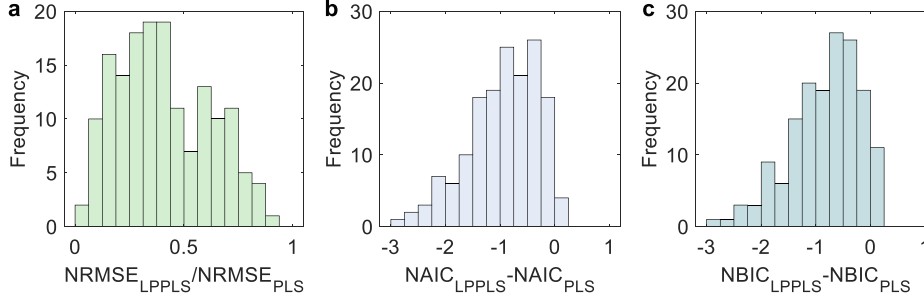

**Fig. 4 | Histograms showing a comparison of the evaluation metrics between the LPPLS and PLS models in fitting 109 geohazard events (with 160 time series monitoring data in total). a** NRMSE ratio, **b** NAIC difference, and (**c**) NBIC difference between the LPPLS and PLS models.

with great potential to mitigate geohazard risks and enhance existing early warning systems. The LPPLS signatures observed in various geohazards imply the presence of a hierarchy of characteristic time scales in the rupture of heterogeneous geomaterials, which provide important insights into the underlying systems and/or physics as well as would be very useful for prediction purposes[21]. More specifically, the unified LPPLS model stands out as a superior approach because it leverages the log-periodic oscillatory structure of acceleration and deceleration phases, transforming them from traditionally perceived nuisances (or noises) into essential components of the predictive framework. By "locking" into this intermittent pattern within the data, this unified failure model enables a deeper understanding of the ongoing damage process and provides a more reliable estimate of the critical time of failure. This innovative use of the log-periodic oscillatory behavior enhances its ability to assess and predict rupture phenomena more accurately than traditional methods. Our future work will focus on implementing this unified failure model for prospective forecast of catastrophic events. Furthermore, LPPLS signatures could serve as indicators for early warning, a concept proven effective for forecasting financial crises[58] and to be explored for geohazards in our future research. Building on the LPPLS model, displacement or velocity thresholds (similar to those developed based on the PLS model[59,60]) might also be developed to facilitate early warnings for imminent catastrophic events.

## Methods
### Time-to-failure models
The conventional power law singularity (PLS) model for time-to-failure analysis[9–11,13] is given by (see Supplementary Note 1 for derivation details):

$$\Omega(t) = A + B(t_c - t)^m, \text{ with } m < 1 \quad (1)$$

where $\Omega$ is an observable quantity (e.g., displacement, strain, energy release, earthquake count, gas emission), $t$ is time, $t_c$ is the time of failure, $A$ and $B$ are constants, and $m$ is a critical exponent. For $0 < m < 1$, $\dot{\Omega}$ diverges at $t_c$ but $\Omega$ converges to the finite value $A$; for $m < 0$, both $\dot{\Omega}$ and $\Omega$ diverge at $t_c$. The parameter set of the PLS model denoted as $\theta_{PLS} = \{A, B, t_c, m\}$ includes four parameters. Parameter $m$ controls the strength of the finite-time singularity (the acceleration of $\Omega$ as $t$ approaches $t_c$) and is determined by the intensity of nonlinearity governing the rate of change in the system's behavior as it approaches failure. Parameter $t_c$ represents the time around which the system transitions from the quasi-static regime (characterized by progressive damage accumulation) to the elasto-dynamical regime (marked by runaway rupture and controlled by the finiteness of the speed of elastic waves). This critical transition is signified by the divergence of $\dot{\Omega}$, while $\Omega$ either culminates at a finite value for $0 < m < 1$ or diverges to infinity for $m < 0$. Beyond $t_c$ (i.e., in the elasto-dynamical regime), the rupture velocity becomes very large and is constrained only by the speed of elastic waves, such that $\Omega$ would approach infinity (even though for $0 < m < 1$ it is finite at $t_c$). For $0 < m < 1$, parameter $A$ is the final value of $\Omega$ at $t_c$ ($t = t_c$), while for $m < 0$, $A$ depends on the initial value of $\Omega$ at time far away from $t_c$ ($t \ll t_c$). Parameter $B$ characterizes the magnitude of the system's response to the approach of failure. Various mechanisms have also been proposed to

explain the micromechanical origin of power law time-to-failure dynamics, such as rate-state friction[61–63] and stress corrosion damage[15,64–66].

We now consider a generalized scenario where the critical exponent is extended from real to complex numbers. Just as the generalization of dimensions from integers to real values gives rise to fractal geometry, enabling the description of non-smooth structures across all scales, the extension of exponents from real to complex numbers reflects the potential existence of discrete hierarchies of scales that punctuate dynamic processes. Then, the first-order Fourier expansion of the general solution for $\Omega$ leads to the following log-periodic power law singularity (LPPLS) model[23,26] (see Supplementary Note 2 for derivation details):

$$\Omega(t) = A + \{B + C\cos[\omega\ln(t_c - t) - \phi]\}(t_c - t)^m, \text{ with } m < 1 \quad (2)$$

expressing a log-periodic correction to the power law scaling, where $\omega$ is the angular log-periodic frequency, $\phi$ is a phase shift, and $C$ is a constant (note that $C = 0$ recovers the simple power law). The parameter set of the LPPLS model $\theta_{LPPLS} = \{A, B, C, t_c, m, \omega, \phi\}$ includes seven parameters. Compared to the parameter set of the PLS model, the three additional parameters in the LPPLS model—$C$, $\omega$, and $\phi$—characterizes the amplitude, angular frequency, and phase offset of the system's log-periodic oscillatory behavior as it approaches failure. Parameter $\omega$ determines the fundamental scaling factor $\lambda$ through $\omega = 2\pi/\ln\lambda$ (see Supplementary Note 2), where the observable remains scale invariant in a discrete manner under scaling of $t_c - t$ by integer powers of $\lambda$. In other words, $\omega$ encapsulates the presence of the symmetry of discrete scale invariance in the function $\Omega(t)$. Parameter $\phi$ defines a characteristic time scale $\rho = \exp(\phi/\omega)$, such that $\cos[\omega\ln(t_c - t) - \phi] = \cos\{\omega\ln[(t_c - t)/\rho]\}$. The relative amplitude of log-periodic oscillations is quantified by $C/B$, typically[26] on the order of $10^{-1}$, and depends on the heterogeneity of the system. The greater the system's heterogeneity, the stronger the log-periodicity[67]. Incorporating a discrete hierarchy of time scales (see Supplementary Note 2), the LPPLS model captures the non-monotonous dynamics of the system with a geometric increase in burst frequency on the approach to $t_c$. A physical mechanism for this log-periodicity relies on the cascade of Mullins-Sekerka instabilities of competing growing cracks[68], driven by the localized and threshold nature of the mechanics of rupture in heterogeneous materials[21,24]. Other possible physical mechanisms include the interplay of inertia, damage, and healing[69,70] as well as the interplay between seismic stress drop and interseismic stress corrosion[71]. The LPPLS model is applicable to both constant and varying loading conditions, as demonstrated in the previous experimental and numerical studies[22,24,72–74].

### Goodness-of-fit tests
We employ a set of evaluation metrics to examine the performance of the LPPLS-based unified failure model versus the PLS-based conventional failure model, including normalized root mean square error (NRMSE), normalized Akaike information criterion (NAIC) and normalized Bayesian information criterion (NBIC) as well as $p$-values derived from the Wilks likelihood-ratio test, the two-sample Kolmogorov-Smirnov (KS) test, and the two-sample Anderson-Darling (AD) test.

For a time series of $N$ measurements of the observable quantity $\Omega = \{\Omega_1, \Omega_2, ..., \Omega_N\}$ recorded at time $\mathbf{t} = \{t_1, t_2, ..., t_N\} \in [\tau, T]$, we compute NRMSE as:

$$\text{NRMSE} = \frac{1}{\Omega_N - \Omega_1} \sqrt{\frac{1}{N} F(\boldsymbol{\theta}; \boldsymbol{\Omega}, \mathbf{t})} \quad (3)$$

where $F$ is the sum of squared errors; refer to equations (S14) and (S24) in the Supplementary Information for the LPPLS and PLS models, respectively.

We test the null hypothesis $H_0$ that the observable quantity $\boldsymbol{\Omega}(\mathbf{t})$ follows the PLS model against the alternative that it follows the LPPLS model. Note that the PLS model is a special case of (or a model nested in) the LPPLS model (i.e., the latter reduces to the former if $C = 0$ or in other words $\boldsymbol{\theta}_{\text{PLS}}$ is a subset of $\boldsymbol{\theta}_{\text{LPPLS}}$). Thus, we use the Wilks likelihood ratio test which is powerful and widely applicable, especially for nested models, because it relies on comparing the likelihoods of the two models, which is a robust way of assessing model fit. It has desirable properties under certain conditions, such as asymptotically following a chi-squared distribution under the null hypothesis, which makes it convenient for deriving $p$-values and making decisions about model selection. The corresponding likelihood-ratio test statistic is given as:

$$\Lambda = -2[\ln L(\hat{\boldsymbol{\theta}}_{\text{PLS}}; \boldsymbol{\Omega}, \mathbf{t}) - \ln L(\hat{\boldsymbol{\theta}}_{\text{LPPLS}}; \boldsymbol{\Omega}, \mathbf{t})] = N \ln \frac{F(\hat{\boldsymbol{\theta}}_{\text{PLS}}; \boldsymbol{\Omega}, \mathbf{t})}{F(\hat{\boldsymbol{\theta}}_{\text{LPPLS}}; \boldsymbol{\Omega}, \mathbf{t})} \quad (4)$$

where $\ln L(\hat{\boldsymbol{\theta}}; \boldsymbol{\Omega}, \mathbf{t})$ is the log-likelihood function (Supplementary Note 7). The test statistic $\Lambda$ converges asymptotically to a chi-squared distribution (with $\kappa = 3$ degrees of freedom, which is the difference in the number of parameters between the LPPLS and PLS models), if $H_0$ happens to be true. For the finite sample here with $N$ number of data points, the distribution of $\Lambda$ is unknown. Here, we perform Monte Carlo simulations (1000 runs) to estimate the $p$-value of the null hypothesis as the fraction of exceedances of the test statistic of simulated data compared to that of the actual data. If the $p$-value is smaller than the prescribed significance level (e.g., 0.05), the null hypothesis $H_0$ is rejected.

We also perform a two-sample KS test and AD test on the null hypothesis $\mathscr{H}_0$ that the probability distributions of the LPPLS and PLS residuals do not differ. If the $p$-value is smaller than the prescribed significance level, the null hypothesis $\mathscr{H}_0$ is rejected, meaning that the residuals of the LPPLS and PLS models are not from the same distribution.

Furthermore, we estimate the relative model quality based on the NAIC and NBIC values calculated as[75]:

$$\text{NAIC} = \frac{2\kappa - 2\ln L(\hat{\boldsymbol{\theta}}; \boldsymbol{\Omega}, \mathbf{t})}{N} = \frac{2\kappa}{N} + \ln F(\hat{\boldsymbol{\theta}}; \boldsymbol{\Omega}, \mathbf{t}) + \ln\left(\frac{2\pi}{N}\right) + 1 \quad (5)$$

$$\text{NBIC} = \frac{\kappa \ln N - 2\ln L(\hat{\boldsymbol{\theta}}; \boldsymbol{\Omega}, \mathbf{t})}{N} = \frac{\kappa \ln N}{N} + \ln F(\hat{\boldsymbol{\theta}}; \boldsymbol{\Omega}, \mathbf{t}) + \ln\left(\frac{2\pi}{N}\right) + 1 \quad (6)$$

which reward the goodness-of-fit while introducing a penalty term for the number of parameters. By employing a comprehensive suite of well-established evaluation metrics—including NAIC, NBIC, Wilks, KS, and AD tests—we ensure a balanced and rigorous assessment of model performance. This broad methodological approach prevents reliance on any single criterion, effectively mitigating potential biases and reinforcing the robustness and objectivity of our evaluation.

## Data availability
The data underlying our study are either from prior published studies or from public databases, with sources provided in the Supplementary Information.

## Code availability
The computer code supporting this study is available for download at Zenodo: https://doi.org/10.5281/zenodo.15362581.

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

## Acknowledgements

Q.L. is grateful for the support by the Swiss National Science Foundation (Grant No. 189882) and the National Natural Science Foundation of China (Grant No. 41961134032), and also acknowledges the National Academic Infrastructure for Supercomputing in Sweden (NAISS), partially funded by the Swedish Research Council through grant agreement no. 2022-06725, for awarding this project access to the LUMI supercomputer, owned by the EuroHPC Joint Undertaking and hosted by CSC (Finland) and the LUMI consortium. D.S. acknowledges partial support from the National Natural Science Foundation of China (Grant No. U2039202, T2350710802), from the Shenzhen Science and Technology Innovation Commission (Grant No. GJHZ20210705141805017) and the Center for Computational Science and Engineering at the Southern University of Science and Technology.

## Author contributions

Q.L. and D.S. designed the research; Q.L. conducted the research; Q.L. and D.S. analyzed the results; Q.L. wrote the manuscript; D.S. reviewed and edited the manuscript.

## Funding

## Competing interests

The authors declare no competing interests.
