## [Transparent Peer Review file · Communications Earth & Environment]

Unified failure model for landslides, rockbursts, glaciers, and volcanoes

Corresponding Author: Dr Qinghua Lei

Version 0:

Decision Letter:

Dear Dr Lei,

Your manuscript titled "Unified failure law for landslides, rockbursts, glaciers, and volcanoes" has now been seen by 2 reviewers, whose comments are appended below. You will see that they find your work of some potential interest. However, they have raised quite substantial concerns that must be addressed. In light of these comments, we cannot accept the manuscript for publication, but would be interested in considering a revised version that fully addresses these serious concerns.

In particular:

- 1) Consider carefully the claims of a unified law and the criticism by both reviewers with respect to this claim. Any claims to such generality must be clearly supported by evidence in the manuscript before publication can be considered.
- 2) Consider carefully the statistical methods applied in the manuscript and present clear justification.

We hope you will find the reviewers' comments useful as you decide how to proceed. Should additional work allow you to address these criticisms, we would be happy to look at a substantially revised manuscript. If you choose to take up this option, please either highlight all changes in the manuscript text file, or provide a list of the changes to the manuscript with your responses to the reviewers.

When resubmitting, please provide a point-by-point response to the reviewers' comments. Please submit your responses as a separate file, distinct from your cover letter where you can add responses to the Editors' comments that you do not want to be made available to the reviewers. Word files are preferred. We recommend that any figures, tables or graphs that are included in the response to reviewers are also included in the main article or Supplementary Information.

If the revision process takes significantly longer than three months, we will be happy to reconsider your paper at a later date, as long as nothing similar has been accepted for publication at Communications Earth & Environment or published elsewhere in the meantime.

Please use the following link to submit your revised manuscript, point-by-point response to the reviewers' comments with a list of your changes to the manuscript text (which should be in a separate document to any cover letter), a tracked-changes version of the manuscript (as a PDF file) and any completed checklist:

Link Redacted

Please do not hesitate to contact us if you have any questions or would like to discuss the required revisions further. Thank you for the opportunity to review your work.

Best regards,

Dr Jan Dettmer
Editorial Board Member
Communications Earth & Environment
orcid.org/0000-0001-8906-8156

Joe Aslin
Deputy Editor
Communications Earth & Environment

EDITORIAL POLICIES AND FORMAT

If you decide to resubmit your paper, please ensure that your manuscript complies with our editorial policies and complete and upload the checklist below as a Related Manuscript file type with the revised article:

Editorial Policy Policy requirements
(Download the link to your computer as a PDF.)

- Behavioural and social science
- Ecological, evolutionary & environmental sciences
- Life sciences

<https://www.nature.com/documents/nr-reporting-summary.zip>

For your information, you can find some guidance regarding format requirements summarized on the following checklist: (<https://www.nature.com/documents/commsj-phys-style-formatting-checklist-article.pdf>) and formatting guide (<https://www.nature.com/documents/commsj-phys-style-formatting-guide-accept.pdf>).

REVIEWER COMMENTS:

Reviewer #1 (Remarks to the Author):

This paper shows that the temporal approach to "catastrophic" failure events can be well described by a model that incorporates oscillatory dynamics over a large number of different data sets associated with landslides, rock bursts, glaciers and volcanoes. In particular, this model statistically outperforms (for the chosen measures) the most conventional time-to-failure model, which indicates a monotonic dynamics. Both models have been around for quite some time. From my perspective, the main novelty of the presented findings is the systematic comparison of the model performances across a large number of diverse data sets outside a lab environment. This is a laudable endeavour, which warrants publication. Having said that, I believe the paper is overselling some aspects and I have a number of related comments I would like to see addressed before final acceptance.

Specifically, the notion of a "unified law" seems too strong. I do not see a clear basis to elevate the LPPLS model to a law. This is because the free parameters of the model need to be individually "fitted" for each data set and there is no clear physical basis what determines them and why in some cases the behaviour is indeed non-monotonic but not in others. The statistical comparison is also largely (if not entirely) relative between the two models. Similarly, I would like to emphasize that there are different ways to penalize having more parameters in one model compared to the other, and the specific choices can lead to a bias one way or the other. Finally, looking at the current evidence from controlled lab experiments for the validity of the model in the literature it is not clear (and I don't see a broad consensus) that the LPPLS model is a better description for many or all cases of material failure (see, for example, very recent lab experiments by Thomas Goebel and collaborators and Jerome Weiss and collaborators). So, I strongly suggest dropping the word "law" (and use something like "description", "relation" or "model" instead throughout) and be more cautious of not overselling your findings.

Reviewer #2 (Remarks to the Author):

The paper proposes a unified framework to predict the temporal occurrence of various catastrophic georuptures, including landslides, glacier ruptures, rockfalls, and volcanic eruptions. Despite their apparent differences, all these natural hazards share a common origin: they result from rupture processes in heterogeneous natural materials. The study demonstrates that these phenomena follow a consistent log-periodic power law singularity leading up to their final breakdown.

While the underlying log-periodic law is not new—it was first discovered and studied in 1996 and successfully applied to individual cases, particularly glacier ruptures—the authors provide a significant advancement by applying a robust calibration method and an extensive array of statistical tests. These results demonstrate that the Log-Periodic Power Law Singularity (LPPLS) model achieves superior predictive accuracy compared to the conventional Power Law Singularity (PLS) approach.

The authors underscore the advantages of LPPLS in capturing the oscillatory behavior (acceleration and deceleration phases) characteristic of rupture dynamics. This oscillatory behavior reflects the physical processes underlying these georuptures, offering additional interpretive insights into the data. Furthermore, while LPPLS had been applied previously to specific scenarios, it was not systematically utilized across different types of georuptures, as proposed in this study. The study's findings are supported by rigorous testing on over 100 georuptures of various types, ranging from glacier collapses and rockfalls to volcanic eruptions. These results highlight the versatility and predictive power of the LPPLS model, paving the way for further applications, particularly in the development of Early Warning Systems for natural hazards.

This paper addresses significant scientific questions that are highly relevant to the field. I found this study to be exceptionally engaging, with remarkable, convincing and noteworthy results. The manuscript is well-written, logically organized, and meticulously structured. In my opinion, this excellent piece of work merits publication in *Nature Communications Earth & Environment*, provided that the authors address and clarify a few minor points.

General comments

First, it is important to acknowledge the extensive and meticulous work undertaken in this study. Heterogeneous data originating from 106 events of diverse nature were systematically retrieved, compiled, and calibrated using the LPPLS (Log-Periodic Power Law Singularity) and LPS (Log-Periodic Singularity) models. For each event, the performance of the respective fits was carefully evaluated, and detailed comparisons were made between the two laws.

The results are compelling. The method developed to compare the performance of these two models is robust and offers valuable insights into their relative efficacy. The systematic approach ensures that the evaluation is consistent and comprehensive, allowing for meaningful conclusions about the strengths and limitations of each model in capturing the dynamics of the events under study.

Calibration

The core of the paper is centered on the calibration of the LPPLS and LPS laws. While I am not a specialist in nonlinear optimization and minimization problems, I found it somewhat challenging to fully understand the detailed steps of the procedure.

In the LPPLS calibration process, my understanding is that you enslaved the linear parameters to the nonlinear parameters (treating the linear parameters as fixed values during certain stages) to reduce the number of variables and simplify the optimization. This iterative procedure is described and reproduced up to equation (S23).

However, the practical implementation of the calibration process remains unclear to me. What is the precise sequence of steps when applying the method? For instance, does one start with equation (S23) and then proceed to equation (S21)? While I assume the code clarifies the process, the steps are not immediately obvious when reading notes S4 or S5. A clearer explanation or additional guidance on the initial steps and their logical flow would be highly beneficial for readers unfamiliar with the specifics of this procedure.

The interpretation could be further developed, as there is currently little to no discussion on the comparison of parameter values obtained for different events, both within each category and between different categories. While I understand that this may not fall within the primary scope of the paper, these results—particularly the parameters derived from the different fits—could serve as a valuable source of discussion and insight.

In particular, there seems to be a lack of analysis regarding the parameter values obtained for the two laws, especially when considering variations in the parameters for the same natural phenomenon. While the primary aim of the paper is to demonstrate that the LPPLS model achieves better predictive performance than the LPS model, the parameters themselves appear to hold significant potential for interpretation.

For example, why are there discrepancies in the parameters obtained for certain glaciers or landslides? A specific point of interest is the occurrence of $m > 0$ and $m < 0$ for the same site, either between different events or across different measurement points. This is particularly notable in locations such as Kagemori, Puigcercos, the gold mine, Vajont, and Merapi. These discrepancies may reflect underlying physical processes, variations in local conditions, or differences in measurement techniques, and a deeper exploration of these aspects would greatly enrich the study.

A discussion addressing these parameter variations and their potential implications could provide a broader understanding of the natural phenomena under investigation and reveal new avenues for future research.

Early Warning Perspectives

You stated that LPPLS enhances existing early warning systems. How? The proposed laws provide a reliable estimate of t_c , the critical time. However, it is important to clarify that t_c does not necessarily correspond to the exact moment of rupture.

- For $m < 0$:

In these cases, t_c signifies the point at which displacement becomes theoretically infinite. However, catastrophic rupture would occur before t_c , as the material would fail at a finite displacement. To enhance early warning capabilities, it is crucial to establish a threshold for the maximum displacement beyond which catastrophic rupture is imminent.

- For $m > 0$:

Here, t_c represents the time at which velocity becomes theoretically infinite, but with a finite displacement. This raises an

important question: does t_c in these cases truly indicate catastrophic rupture? The relationship between velocity-driven predictions and the actual occurrence of rupture needs further clarification, particularly in scenarios where the displacement may remain below critical thresholds.

These distinctions between $m > 0$ and $m < 0$ cases highlight the need for additional investigation into the physical meaning of the parameters and their implications for early warning systems.

The concept of early warning also raises critical questions about the temporal scope of instability assessment. Specifically, how far in advance would it be possible to reliably detect and assess the onset of instability? Additionally, how long must the time series be to perform a robust and reliable fit using the proposed model?

These are essential questions that, while not addressed in detail in this paper, represent exciting avenues for future research. The framework and findings presented here lay the groundwork for exploring such questions and open up wonderful perspectives for advancing early warning systems. Investigating these aspects further could significantly enhance the practical applicability of the model, particularly for real-time monitoring and disaster mitigation efforts.

Physical Significance of m and ω

What is the physical significance of the parameter m ? More specifically, how can velocity become infinite while the corresponding displacement remains finite? Additionally, how can a finite displacement still represent a catastrophic rupture event, such as a landslide or glacier collapse, as presented in the study?

Typically, m is expected to take negative values in many physical models. Could you please elaborate on this point further, particularly in the context of your findings? While the mathematical framework, as demonstrated in Section S2, is clear, the physical interpretation of the parameter m remains less intuitive.

To further clarify, consider the expression $(t_c - t)^m$:

- For $m < 0$: The displacement increases toward infinity as t approaches t_c (the critical time).
- For $m > 0$: The displacement approaches zero as t nears t_c .

Given these relationships, how can we interpret this in terms of physical process? Specifically, if $m > 0$, and displacement tends toward zero, could this still indicate a catastrophic rupture, or is there a need for further criteria or thresholds to define such events?

Additionally, if the system is accelerating toward rupture, as implied by the behavior of m , it seems that certain conditions (represented by parameters A and B) would need to be adjusted or controlled to better predict or understand this acceleration. Could you elaborate on how these parameters influence the interpretation of m and its connection to catastrophic events?

I also observed that in many cases, $\omega = 4.94 \Leftrightarrow \lambda = 3.57$, which is at the lower bound of the interval in the fitting procedure (e.g., for the majority of glaciers). Does this suggest that the fitting process is reaching its limitations in these cases? Furthermore, could the presence of a second harmonic contribute to this behavior? For instance, if a second harmonic were present, it could result in a superposition of $\lambda = 2$ and $\lambda = 4$, potentially influencing the fits and explaining the observed tendency toward the lower bound. This raises intriguing questions about whether the model fully captures the complexity of the dynamics in such scenarios.

Specific Comments

Line 84:

You selected one event per category (landslide, rockburst, glacier, volcanoes) to detail the approach you developed. Why were these specific events chosen? Are they representative of their respective categories, or were they selected because they best illustrate the strengths of your approach? Alternatively, were these the first cases you studied, and if so, what criteria guided their selection? Clarifying this choice would help readers understand how these cases fit into the broader scope of your study.

Line 164:

You present 16 additional cases in semi-detail. Is the purpose of including these cases to illustrate the robustness of your approach across different metrics? Or is it to highlight variations within the same category of events? A brief explanation in the text would help clarify the rationale behind this selection and how it supports the overall findings of the paper.

Table 1 and 2:

On a first reading, it is somewhat difficult to interpret the different parameters listed, as the LPPLS and LPS models are defined only later in the paper. It would be helpful to refer explicitly to the relevant equations in the table captions to guide readers. Additionally, including the definitions of the metrics used in Table 1 would improve clarity and make it easier for readers to follow the discussion of the results.

Line 169, 173, 175:

For extensometer or prism measurements, you work with a time series recorded at a single, fixed point. However, in the case of surface measurements such as those obtained from GBSAR or satellite images, the approach differs due to the spatial nature of the data. How do you determine the specific point to measure within the surface data? Once this point is chosen, what methodology do you use to consistently track it across the entire time series?

For instance, is the point selected based on predefined criteria such as maximum deformation, a specific region of interest, or other geophysical indicators? Furthermore, given the potential for spatial variability in surface measurements, how do you ensure that the selected point remains representative of the broader system or phenomenon being studied?

Line 283-284:

The statement that the LPPLS model "is applicable to both constant and varying loading conditions" raises some questions in the context of natural catastrophic events. My understanding is that the LPPLS model is particularly well-suited for capturing the endogenous development of instabilities under relatively stable conditions.

However, I am less certain about its applicability to scenarios involving varying loading conditions (exogenous forcing), especially when these external influences are not explicitly known or measured. How can such a law, which is inherently designed to describe internal, self-organized dynamics, account for responses to external, time-varying forcing that is undefined or unmeasured?

Communications Earth & Environment is committed to improving transparency in authorship. As part of our efforts in this direction, we are now requesting that all authors identified as 'corresponding author' create and link their Open Researcher and Contributor Identifier (ORCID) with their account on the Manuscript Tracking System prior to acceptance. ORCID helps the scientific community achieve unambiguous attribution of all scholarly contributions. You can create and link your ORCID from the home page of the Manuscript Tracking System by clicking on 'Modify my Springer Nature account' and following the instructions in the link below. Please also inform all co-authors that they can add their ORCIDs to their accounts and that they must do so prior to acceptance.

Version 1:

Decision Letter:

Dear Dr Lei,

Your manuscript titled "Unified failure model for landslides, rockbursts, glaciers, and volcanoes" has now been seen by our reviewers, whose comments appear below. In light of their advice we are delighted to say that we are happy, in principle, to publish a suitably revised version, that discusses insights from recent lab experiments, in Communications Earth & Environment.

We therefore invite you to revise your paper one last time to address the remaining concerns of our reviewers. At the same time we ask that you edit your manuscript to comply with our format requirements and to maximise the accessibility and therefore the impact of your work.

EDITORIAL REQUESTS:

*****Please take care to match our formatting and policy requirements. We will check revised manuscript and return manuscripts that do not comply. Such requests will lead to delays. *****

SUBMISSION INFORMATION:

OPEN ACCESS:

Communications Earth & Environment is a fully open access journal. Articles are made freely accessible on publication. For further information about article processing charges, open access funding, and advice and support from Nature Research, please visit <https://www.nature.com/commsenv/open-access>

Link Redacted

Best regards,

Dr Jan Dettmer
Editorial Board Member
Communications Earth & Environment
orcid.org/0000-0001-8906-8156

Joe Aslin
Deputy Editor,
Communications Earth & Environment
Consulting Editor,
Communications Sustainability

<https://www.nature.com/commsenv/>
Twitter: @CommsEarth

REVIEWERS' COMMENTS:

Reviewer #1 (Remarks to the Author):

I appreciate the changes made by the authors, almost all my concerns have been addressed. Given the now more extended discussions of the physical relevance of the different parameters (including m and α) and the underlying causes in the Discussion section I would like to see a discussion of the insights from the recent controlled lab experiments I mentioned. In addition to the paper the authors cite in their reply (Goebel et al. 2024), I believe the following ones (including some of the references therein) would help to present a more complete picture of our current understanding of failure processes including variations in m and α for the targeted audience:

Phys. Rev. Lett. 122, 015502 (2019)

Phys. Rev. E 108, 014131 (2023)

Once this is addressed, I believe the paper is acceptable for publication.

Reviewer #2 (Remarks to the Author):

I would like to thank the authors for their clear and thoughtful revisions. I have no further comments and sincerely hope this valuable contribution will be accepted for publication in the journal.

Response to Review Comments

In this response document, the reviewers' comments are stated first (in black), followed by our "Replies" (in blue). The line and equation numbers referred in this response document correspond to the uploaded revised manuscript or supporting information with the changed parts highlighted in red.

Reviewer #1 (Remarks to the Author):

This paper shows that the temporal approach to "catastrophic" failure events can be well described by a model that incorporates oscillatory dynamics over a large number of different data sets associated with landslides, rock bursts, glaciers and volcanoes. In particular, this model statistically outperforms (for the chosen measures) the most conventional time-to-failure model, which indicates a monotonic dynamics. Both models have been around for quite some time. From my perspective, the main novelty of the presented findings is the systematic comparison of the model performances across a large number of diverse data sets outside a lab environment. This is a laudable endeavour, which warrants publication. Having said that, I believe the paper is overselling some aspects and I have a number of related comments I would like to see addressed before final acceptance.

Reply: Thank you very much for your thoughtful feedback and for recognizing the value of our systematic model comparison. We appreciate your concerns about potential overstatements and have carefully revised the text to ensure a balanced presentation. Additionally, we have thoroughly addressed your specific comments below to clarify and strengthen our arguments.

Specifically, the notion of a "unified law" seems too strong. I do not see a clear basis to elevate the LPPLS model to a law. This is because the free parameters of the model need to be individually "fitted" for each data set and there is no clear physical basis what determines them and why in some cases the behaviour is indeed non-monotonic but not in others. The statistical comparison is also largely (if not entirely) relative between the two models. Similarly, I would like to emphasize that there are different ways to penalize having more parameters in one model compared to the other, and the specific choices can lead to a bias one way or the other. Finally, looking at the current evidence from controlled lab experiments for the validity of the model in the literature it is not clear (and I don't see a broad consensus) that the LPPLS model is a better description for many or all cases of material failure (see, for example, very recent lab experiments by Thomas Goebel and collaborators and Jerome Weiss and collaborators). So, I strongly suggest dropping the word "law" (and use something like "description", "relation" or "model" instead throughout) and be more cautious of not overselling your findings.

Reply: Thank you for your detailed and constructive comments. We understand your concerns regarding the use of "unified law" in the paper and have carefully revised the manuscript to adopt a more neutral tone, replacing "law" with "model" where appropriate.

We acknowledge that the model parameters are obtained through fitting with observational data. However, each parameter has a well-defined physical meaning and is linked to the underlying processes that govern geomaterial ruptures. Our physics-based model not only captures the fundamental processes governing geomaterial rupture phenomena but also remains flexible enough for calibration against monitoring data. Each system is inherently unique, with its own heterogeneity, damage and healing history, and external conditions. Our model adapts to these specific characteristics and idiosyncratic patterns, making this adaptability an extraordinary strength of our framework. We have better clarified this point in the revised manuscript (see Lines 79-80). In addition, we have provided detailed descriptions about the physical meanings of all the parameters in the PLS and LPPLS models and also explained their connections

to the underlying physical processes of material damage and rupture (see Lines 300-314, Lines 326-335, and Lines 340-342).

No single criterion is perfect, as each tends to be sensitive to different aspects of the data and the model. Therefore, the optimal approach is to rely on widely accepted, standard criteria and to utilize a comprehensive set covering diverse characteristics. This is precisely the approach we have adopted. By employing a comprehensive suite of well-established evaluation metrics—including NAIC, NBIC, Wilks, KS, and AD tests—we ensure a balanced and rigorous assessment of model performance. This broad methodological approach prevents reliance on any single criterion, effectively mitigating potential biases and reinforcing the robustness and objectivity of our evaluation. This combination of assessments yields consistent results, collectively reinforcing the robustness of our conclusion. We have better clarified this point in the revised manuscript (see Lines 381-385).

Thanks for highlighting the recent laboratory studies by Thomas Goebel and Jerome Weiss and their collaborators. While their experimental data on rock sample ruptures appear relatively smooth, some intermittency is still evident; for example, see Fig. 1 in Goebel et al. (2024). We agree that, in many cases, including some presented in our paper, log-periodicity is not visually significant. This may be because the heterogeneity in these systems is relatively weak, as observed especially for glacier cases. However, our quantitative model comparisons and rigorous statistical analyses provide clear evidence that the LPPLS model consistently outperforms the PLS model. We emphasize that a lack of visually apparent log-periodic oscillations does not imply mathematical or statistical insignificance. We have better clarified this point in the revised manuscript (see Lines 246-250).

References:

Goebel, T. H. W., Schuster, V., Kwiatek, G., Pandey, K. & Dresen, G. A laboratory perspective on accelerating preparatory processes before earthquakes and implications for foreshock detectability. *Nat Commun* 15, 5588 (2024).

Reviewer #2 (Remarks to the Author):

The paper proposes a unified framework to predict the temporal occurrence of various catastrophic georuptures, including landslides, glacier ruptures, rockfalls, and volcanic eruptions. Despite their apparent differences, all these natural hazards share a common origin: they result from rupture processes in heterogeneous natural materials. The study demonstrates that these phenomena follow a consistent log-periodic power law singularity leading up to their final breakdown.

While the underlying log-periodic law is not new—it was first discovered and studied in 1996 and successfully applied to individual cases, particularly glacier ruptures—the authors provide a significant advancement by applying a robust calibration method and an extensive array of statistical tests. These results demonstrate that the Log-Periodic Power Law Singularity (LPPLS) model achieves superior predictive accuracy compared to the conventional Power Law Singularity (PLS) approach.

The authors underscore the advantages of LPPLS in capturing the oscillatory behavior (acceleration and deceleration phases) characteristic of rupture dynamics. This oscillatory behavior reflects the physical processes underlying these georuptures, offering additional interpretive insights into the data. Furthermore, while LPPLS had been applied previously to specific scenarios, it was not systematically utilized across different types of georuptures, as proposed in this study.

The study's findings are supported by rigorous testing on over 100 georuptures of various types, ranging from glacier collapses and rockfalls to volcanic eruptions. These results highlight the versatility and predictive

power of the LPPLS model, paving the way for further applications, particularly in the development of Early Warning Systems for natural hazards.

This paper addresses significant scientific questions that are highly relevant to the field. I found this study to be exceptionally engaging, with remarkable, convincing and noteworthy results. The manuscript is well-written, logically organized, and meticulously structured. In my opinion, this excellent piece of work merits publication in Nature Communications Earth & Environment, provided that the authors address and clarify a few minor points.

Reply: We sincerely appreciate your thoughtful and detailed review of our manuscript. We are grateful for your recognition of our work, especially your comments on the broad applicability of the LPPLS model. We have carefully addressed the points you raised and have included the necessary clarifications to further strengthen our manuscript. Thank you again for your valuable feedback and for supporting our study for publication.

General comments

First, it is important to acknowledge the extensive and meticulous work undertaken in this study. Heterogeneous data originating from 106 events of diverse nature were systematically retrieved, compiled, and calibrated using the LPPLS (Log-Periodic Power Law Singularity) and LPS (Log-Periodic Singularity) models. For each event, the performance of the respective fits was carefully evaluated, and detailed comparisons were made between the two laws.

The results are compelling. The method developed to compare the performance of these two models is robust and offers valuable insights into their relative efficacy. The systematic approach ensures that the evaluation is consistent and comprehensive, allowing for meaningful conclusions about the strengths and limitations of each model in capturing the dynamics of the events under study.

Calibration

The core of the paper is centered on the calibration of the LPPLS and LPS laws. While I am not a specialist in nonlinear optimization and minimization problems, I found it somewhat challenging to fully understand the detailed steps of the procedure.

In the LPPLS calibration process, my understanding is that you enslaved the linear parameters to the nonlinear parameters (treating the linear parameters as fixed values during certain stages) to reduce the number of variables and simplify the optimization. This iterative procedure is described and reproduced up to equation (S23).

However, the practical implementation of the calibration process remains unclear to me. What is the precise sequence of steps when applying the method? For instance, does one start with equation (S23) and then proceed to equation (S21)? While I assume the code clarifies the process, the steps are not immediately obvious when reading notes S4 or S5. A clearer explanation or additional guidance on the initial steps and their logical flow would be highly beneficial for readers unfamiliar with the specifics of this procedure.

Reply: Thank you very much for raising this point. We have now provided a detailed description of the implementation steps for the LPPLS and PLS model calibration in the Supplementary Notes S4 and S5 (see Lines 147-161 and Lines 191-205 in the Supplementary Information). This explanation, together with the code to be shared upon the paper acceptance, will help readers better understand our calibration algorithm and procedure.

The interpretation could be further developed, as there is currently little to no discussion on the comparison of

parameter values obtained for different events, both within each category and between different categories. While I understand that this may not fall within the primary scope of the paper, these results—particularly the parameters derived from the different fits—could serve as a valuable source of discussion and insight.

In particular, there seems to be a lack of analysis regarding the parameter values obtained for the two laws, especially when considering variations in the parameters for the same natural phenomenon. While the primary aim of the paper is to demonstrate that the LPPLS model achieves better predictive performance than the LPS model, the parameters themselves appear to hold significant potential for interpretation.

For example, why are there discrepancies in the parameters obtained for certain glaciers or landslides? A specific point of interest is the occurrence of $m > 0$ and $m < 0$ for the same site, either between different events or across different measurement points. This is particularly notable in locations such as Kagemori, Puigcercos, the gold mine, Vajont, and Merapi. These discrepancies may reflect underlying physical processes, variations in local conditions, or differences in measurement techniques, and a deeper exploration of these aspects would greatly enrich the study.

A discussion addressing these parameter variations and their potential implications could provide a broader understanding of the natural phenomena under investigation and reveal new avenues for future research.

Reply: Thank you for this very constructive comment. We agree that providing some interpretation to the variation of the parameters derived could be very valuable. Yes, the variation of m across different instruments placed at different locations for the same sites may reflect the inherent heterogeneity of natural systems or the potential influence of different underlying mechanisms. The variation of m across different types of instruments for the same site may be attributed to the fact that they reflect distinct processes operating within the same system. Additionally, the m values for failure events occurring at different times at the same site can also change, highlighting the nonstationary nature of natural systems. We have added a thorough discussion on these parameter variations and their potential implications in the revised manuscript (see Lines 259-268).

Early Warning Perspectives

You stated that LPPLS enhances existing early warning systems. How? The proposed laws provide a reliable estimate of t_c , the critical time. However, it is important to clarify that t_c does not necessarily correspond to the exact moment of rupture.

- For $m < 0$:

In these cases, t_c signifies the point at which displacement becomes theoretically infinite. However, catastrophic rupture would occur before t_c , as the material would fail at a finite displacement. To enhance early warning capabilities, it is crucial to establish a threshold for the maximum displacement beyond which catastrophic rupture is imminent.

- For $m > 0$:

Here, t_c represents the time at which velocity becomes theoretically infinite, but with a finite displacement. This raises an important question: does t_c in these cases truly indicate catastrophic rupture? The relationship between velocity-driven predictions and the actual occurrence of rupture needs further clarification, particularly in scenarios where the displacement may remain below critical thresholds.

These distinctions between $m > 0$ and $m < 0$ cases highlight the need for additional investigation into the physical meaning of the parameters and their implications for early warning systems.

The concept of early warning also raises critical questions about the temporal scope of instability assessment. Specifically, how far in advance would it be possible to reliably detect and assess the onset of instability? Additionally, how long must the time series be to perform a robust and reliable fit using the proposed

model?

These are essential questions that, while not addressed in detail in this paper, represent exciting avenues for future research. The framework and findings presented here lay the groundwork for exploring such questions and open up wonderful perspectives for advancing early warning systems. Investigating these aspects further could significantly enhance the practical applicability of the model, particularly for real-time monitoring and disaster mitigation efforts.

Reply: Thank you for offering this very insightful comment. Yes, rigorously speaking, the time t_c does not correspond to the exact moment of final rupture. We emphasize that the finite-time singularity at t_c is nothing but the signature of a critical transition from one regime to another. More specifically, t_c represents the time around which the system transitions from the quasi-static regime (characterised by progressive damage accumulation) to the elasto-dynamical regime (marked by runaway rupture and controlled by the finiteness of the speed of elastic waves). Given that the rupture during the elasto-dynamical regime has a very short duration, t_c is considered to be a good indicator of the final failure time. Note that this critical transition is signified by the divergence of velocity irrespective of the m value. At t_c , the displacement either culminates at a finite value (for $0 < m < 1$) or diverges to infinity for (for $m < 0$). Beyond t_c , the rupture velocity in the elasto-dynamical regime becomes constrained only by the speed of elastic waves, such that the displacement will also reach very large values (even if it is finite at t_c for $0 < m < 1$). Thus, the finite value of displacement at t_c for $0 < m < 1$ simply means that the system transitions into the elasto-dynamical regime when the displacement is still finite. We have better clarified this point in the revised manuscript (see Lines 302-309). We fully agree that it is important to highlight in the current paper the potential for further application of our model for early warning purposes. We have added some discussion on this point in the revised manuscript (see Lines 287-291).

Physical Significance of m and ω

What is the physical significance of the parameter m ? More specifically, how can velocity become infinite while the corresponding displacement remains finite? Additionally, how can a finite displacement still represent a catastrophic rupture event, such as a landslide or glacier collapse, as presented in the study?

Typically, m is expected to take negative values in many physical models. Could you please elaborate on this point further, particularly in the context of your findings? While the mathematical framework, as demonstrated in Section S2, is clear, the physical interpretation of the parameter m remains less intuitive.

To further clarify, consider the expression $(t_c - t)^m$:

- For $m < 0$: The displacement increases toward infinity as t approaches t_c (the critical time).
- For $m > 0$: The displacement approaches zero as t nears t_c .

Given these relationships, how can we interpret this in terms of physical process? Specifically, if $m > 0$, and displacement tends toward zero, could this still indicate a catastrophic rupture, or is there a need for further criteria or thresholds to define such events?

Additionally, if the system is accelerating toward rupture, as implied by the behavior of m , it seems that certain conditions (represented by parameters A and B) would need to be adjusted or controlled to better predict or understand this acceleration. Could you elaborate on how these parameters influence the interpretation of m and its connection to catastrophic events?

Reply: Thank you for this insightful comment. As replied to your former comment, the finite-time singularity at t_c is nothing but the signature of the system's transition from the the quasi-static regime (characterised by progressive damage accumulation) to the elasto-dynamical regime (marked by runaway rupture and controlled by the finiteness of the speed of elastic waves). This critical transition is signified by

the divergence of velocity irrespective of the m value. At t_c , the displacement either culminates at a finite value (for $0 < m < 1$) or diverges to infinity for (for $m < 0$). Thus, for $0 < m < 1$, the displacement does not approach zero at t_c as stated in your comment; instead, it accelerates and culminates at the finite value A at t_c at which time the velocity diverges (physically it accelerates to the elasto-dynamic regimes limited by the compressive or shear wave velocity). Beyond t_c , the rupture velocity in the elasto-dynamical regime becomes constrained only by the speed of elastic waves, such that the displacement will grow to very large values (even if it is finite at t_c for $0 < m < 1$). Thus, the finite value of displacement at t_c for $0 < m < 1$ simply means that the system transitions into the elasto-dynamical regime when the displacement is still finite. We have better clarified this point in the revised manuscript (see Lines 302-309).

Regarding the m value, given that $m = 1 - 1/(\alpha - 1)$, where α is the exponent in the equation $\ddot{\Omega} = \eta \dot{\Omega}^\alpha$, with $\alpha > 1$ (i.e., equation S1 in the Supplementary Information), we can expect $m < 0$ for $1 < \alpha < 2$ and $0 < m < 1$ for $\alpha > 2$. Previous observational data have indicated that α typically varies around 2, ranging from smaller than 2 to larger than 2 (Voight 1988, 1989; Intrieri et al. 2019). This is consistent with our analysis, as revealed by a newly added figure (see Supplementary Fig. S15) showing the histograms of the m and α values of the PLS and LPPLS models for the 109 geohazard events. The higher α value means the presence of a stronger positive feedback during the acceleration process. Indeed, some physical models suggest that $1 < \alpha < 2$, resulting from the nucleation and propagation of noninteracting subcritical cracks (Kilburn et al. 1998; Kilburn & Petley 2003). We interpret that the occurrence of $\alpha > 2$ may be related to the dominance of crack interaction and coalescence characterized by stronger positive feedbacks. We have added a discussion about the physical meaning of the m and α values in the revised manuscript (see Lines 251-259). In addition, we have emphasized the physical significance of parameter ω in the revised manuscript (see Lines 326-332).

References:

- Intrieri, E., Carlà, T. & Gigli, G. Forecasting the time of failure of landslides at slope-scale: A literature review. *Earth-Sci. Rev.* 193, 333–349 (2019).
- Kilburn, C. R. J. & Petley, D. N. Forecasting giant, catastrophic slope collapse: lessons from Vajont, Northern Italy. *Geomorphology* 54, 21–32 (2003).
- Kilburn, C. R. J. & Voight, B. Slow rock fracture as eruption precursor at Soufriere Hills Volcano, Montserrat. *Geophys. Res. Lett.* 25, 3665–3668 (1998).
- Voight, B. A method for prediction of volcanic eruptions. *Nature* 332, 125–130 (1988).
- Voight, B. A relation to describe rate-dependent material failure. *Science* 243, 200–203 (1989).

Response to Review Comments

In this response document, the reviewers' comments are stated first (in black), followed by our "Replies" (in blue). The line and equation numbers referred in this response document correspond to the uploaded revised manuscript or supporting information with the changed parts highlighted in red.

Reviewer #1 (Remarks to the Author):

I appreciate the changes made by the authors, almost all my concerns have been addressed. Given the now more extended discussions of the physical relevance of the different parameters (including m and α) and the underlying causes in the Discussion section I would like to see a discussion of the insights from the recent controlled lab experiments I mentioned. In addition to the paper the authors cite in their reply (Goebel et al. 2024), I believe the following ones (including some of the references therein) would help to present a more complete picture of our current understanding of failure processes including variations in m and α for the targeted audience:

Phys. Rev. Lett. 122, 015502 (2019)

Phys. Rev. E 108, 014131 (2023)

Once this is addressed, I believe the paper is acceptable for publication.

Reply: Thank you very much for your positive feedback on our revised manuscript. We agree that incorporating a discussion of the insights from recent laboratory experiments would be valuable. We have included such a discussion to explain that the variation of m values across different systems may be attributed to differences in heterogeneity conditions. Extensive laboratory studies have shown that heterogeneities associated with crack distribution, pore space, stress field, and/or fault roughness strongly control the precursory acceleration behaviour of geomaterials as they approach catastrophic failure¹⁻⁶. We have also discussed why the m values for failure events occurring at different times at the same site can change. This is because the system at the onset of each failure cycle may have different initial damage states, which can indeed affect its precursory acceleration pattern, as demonstrated in laboratory experiments^{4,6}. The added texts can be found in Lines 258-271 of the revised manuscript.

References

1. Vasseur, J. *et al.* Heterogeneity: The key to failure forecasting. *Sci. Rep.* **5**, 13259 (2015).
2. Vasseur, J. *et al.* Does an inter-flaw length control the accuracy of rupture forecasting in geological materials? *Earth Planet. Sci. Lett.* **475**, 181–189 (2017).
3. Vu, C.-C., Amitrano, D., Plé, O. & Weiss, J. Compressive failure as a critical transition: Experimental evidence and mapping onto the universality class of depinning. *Phys. Rev. Lett.* **122**, 015502 (2019).
4. Cartwright-Taylor, A. *et al.* Catastrophic failure: how and when? Insights from 4-D in situ X-ray microtomography. *J. Geophys. Res. Solid Earth* **125**, e2020JB019642 (2020).
5. Patton, A., Goebel, T., Kwiatek, G. & Davidsen, J. Large-scale heterogeneities can alter the characteristics of compressive failure and accelerated seismic release. *Phys. Rev. E* **108**, 014131 (2023).
6. Goebel, T. H. W., Schuster, V., Kwiatek, G., Pandey, K. & Dresen, G. A laboratory perspective on accelerating preparatory processes before earthquakes and implications for foreshock detectability. *Nat. Commun.* **15**, 5588 (2024).